# Vitamin D Daily versus Monthly Administration: Bone Turnover and Adipose Tissue Influences

**DOI:** 10.3390/nu10121934

**Published:** 2018-12-06

**Authors:** Luca Dalle Carbonare, Maria Teresa Valenti, Francesco del Forno, Giorgio Piacentini, Angelo Pietrobelli

**Affiliations:** 1Clinic of Internal Medicine, Section D, Department of Medicine, University of Verona, 37134 Verona, Italy; luca.dallecarbonare@univr.it (L.D.C.); francesco.delforno@univr.it (F.d.F.); 2Specialized Regional Center for Biomolecular and Histomorphometric Research for Skeletal and Degenerative Diseases, 37134 Verona, Italy; 3Department of Surgical Sciences, Dentistry, Gynecology and Pediatrics, Pediatric Division, University of Verona, 37134 Verona, Italy; giorgio.piacentini@univr.it (G.P.); angelo.pietrobelli@univr.it (A.P.); 4Pennington Biomedical Research Center, Baton Rouge, LA 70808, USA

**Keywords:** adipose tissue, dosage, waist circumference, monthly, vitamin D

## Abstract

Vitamin D is involved in bone metabolism and in many various extra-skeletal diseases such as malabsorption syndromes, cardiovascular and metabolic diseases, cancer, and autoimmune and neurological diseases. However, data on the optimal route of administration are not consistent. The aims of our study were to analyze not only the influence of daily vs. monthly administration of vitamin D on bone metabolism and bone turnover, but also the effects of different routes of administration on fat mass in a cohort of adults with low levels of 25(OH) vitamin D3 at baseline. We analyzed 44 patients with hypovitaminosis at baseline and after six months of two different regimens of administration: seven drops (1750 IU)/day vs. 50,000 IU/month. We found that the two regimens were equivalent; 36 out of 44 patients reached the normal range of vitamin D after six months of treatment. Interestingly, the main determinant of vitamin D at baseline was the waist circumference. In addition, 22 patients treated by monthly regimen were evaluated after 18 months of treatment. At the end of follow-up, patients showed normal levels of vitamin D, with increased calcium levels and decreased bone turnover. Waist circumference also decreased. Our results support the efficacy of vitamin D3 given monthly both for correcting hypovitaminosis and for maintaining vitamin D levels. The relationship between serum 25(OH)vitamin D3 concentration and waist circumference supports vitamin D having a protective role in the current setting, since waist size is directly associated with the risk of cardiovascular and metabolic diseases.

## 1. Introduction

Vitamin D and its hormonally active metabolite form, (1,25(OH)_2_D) are involved in bone metabolism and in many various extra-skeletal diseases such as malabsorption syndromes, cardiovascular and metabolic diseases, cancer, and autoimmune and neurological diseases [1,2,3]. Vitamin D is an important regulator of runt-related transcription factor 2 (*RUNX2*), the master gene promoting osteogenic differentiation. In fact, the dowregulation of *RUNX2* due to vitamin D promotes osteoblast maturation from mesenchymal stem cells (MSCs) [4]. However, proliferation of MSCs is affected by vitamin D [5] which thereby promotes osteogenic differentiation and reduces MSC proliferation [6]. In addition, MSCs are also the precursor cells of adipocytes and peroxisome proliferator-activated receptor gamma (PPARγ) is the transcription factor essential for adipogenic differentiation [7]. Importantly, reciprocal control between osteogenic and adipogenic differentiation occurs [8,9], and we can suppose that hormonal vitamin D, via inducing osteogenic maturation, could influence adipogenic differentiation.

Despite the accepted role of vitamin D and its pivotal role in the activation process, there is still an ongoing discussion to define the optimal 25(OH) vitamin D3 serum concentration for the maintenance of bone health, as well as what effective vitamin D replacement therapy should involve [2,10,11]. On the other hand, we also have no clear picture of the effects of daily vs. monthly administration and low vs. high dosage [2,12]. As far as we know, in addition to vitamin D trials for skeletal outcomes, the role of Vitamin D in infectious disease is currently a hot topic of investigation [13,14]. However, no information is available for the optimal benefits of vitamin D supplementation regarding different dosages related to time of administration, subsequent influences on clinical outcomes, and mostly importantly, dosing regimen influences on fat mass and bone metabolism [2,12]. The roles of vitamin D in energetic metabolism are well known [15] and many obese children and adults present low blood concentrations of vitamin D [12]. Indeed, a number of observational studies documented a substantial role of vitamin D deficiency in etiopathogenesis of metabolic syndrome and other obesity-related complications. However, we still lack evidence from interventional studies confirming the causal characteristics of these relationships [11], as well as the real relationship between bone mineral content and bone mineral density with fat mass and fat-free mass [13]. Another important point that could be clarified is related to absorption depending on individual metabolic variables influenced by the environment [14,16].

The aims of our study were not only to analyze the influence of daily vs. monthly administration of vitamin D on bone metabolism and bone turnover, but also the influence of different administration on fat mass in a cohort of adults with either low vitamin D status or low baseline concentrations of serum 25(OH) vitamin D3.

In addition, we evaluated the efficacy of a single monthly dose of 50,000 IU of cholecalciferol for maintaining normal values of 25(OH) vitamin D3 after one year of treatment

## 2. Patients and Methods

The study design is reported in Figure 1. All patients were enrolled during routine visits finalized to evaluate the presence of metabolic bone disease in the Specialized Regional Center for Biomolecular and Histomorphometric Research for Skeletal and Degenerative Diseases of the Azienda Universitaria Ospedaliera Integrata di Verona, Italy (AOUI). All patients gave their written informed consent, approved by the local ethical committee of AOUI. The population included 40 patients—20 men and 20 women, aged 77 ± 8 years. The patients showing hypovitaminosis D, defined as a level of 25(OH) vitamin D3 between 10 and 30 ng/mL, were consecutively randomized into two groups as follows: the first group was treated with a daily dose of 1750 IU of cholecalciferol (seven drops/day), while the second group was treated with a monthly single dose of 50,000 IU of cholecalciferol.

All patients enrolled in the study were naive for the consumption of vitamin D3 supplements.

Exclusion criteria for all patients were the presence of independent risk factors for vitamin D and bone metabolism alterations, treatment in the last 24 months with bisphosphonates, teriparatide, denosumab, estrogens, or other drugs influencing bone metabolism and vitamin D status, such as corticosteroids or anticonvulsants, as well as any condition affecting bone metabolism, such as dysthyroidism, hyperparathyroidism, malabsorption, or severe hepatic and renal failure.

All patients underwent two consecutive clinical and biohumoral evaluations at baseline and after six months of treatment. During the clinical evaluations, moderate physical activity (at least a walk of about 30 min twice weekly) was suggested to all patients.

At six months, we selected patients of group 2 (treated with a single monthly dose of 50,000 UI of cholecalciferol), and we maintained supplementation for a further 12 months, after which the patients underwent the same evaluations as at baseline.

### 2.1. Biochemical Evaluations

Based on the single-study protocol, all patients underwent the biochemical evaluations described below.

The serum levels of calcium (s-calcium; normal values 8.41–10.42 mg/dL), s-phosphate (2.63–4.49 mg/dL), s-creatinine (0.49–1.19 mg/dL), s-parathyroid hormone (PTH, 6.5–36.8 pg/mL), s-25(OH) vitamin D3 (≥30 ng/mL), s-alkaline phosphatase (ALP, 50–130 U/L) as a marker of bone formation, and s-C-terminal telopeptide of type 1 collagen (CTX, 0.100–0.700 ng/mL), as a marker of bone resorption, were determined. In particular, serum 25(OH) vitamin D3 was measured using commercially available enzyme immunoassay ELISA kits (Biomedica Medizinprodukte, Vienna, Austria) with a fully automated microplate analyzer Personal LAB (Adaltis, Rome, Italy). The intraassay coefficients of variation (CV) were 6% for 25(OH) vitamin D3 (interassay CV = 9%),

### 2.2. Clinical Evaluations

At baseline, the heights (cm), weights (kg), waist circumferences (cm), and the waist-to-height ratios (WtHR) were collected for all patients. All the anthropometric measurements were made at a standard time of day and with standardized equipment. Body mass index (BMI; kg/m^2^) was also calculated. In addition, a questionnaire on daily calcium intake was collected at baseline. 

### 2.3. Bone Densitometry

In all patients, bone densitometry at the proximal femur was performed at baseline using dual-energy X-ray absorptiometry (DXA; Hologic Discovery 4500 Acclaim, Walton, FL, USA). Daily quality assurance checks were performed using a calibration standard. Patients undergoing monthly treatment (group 2) underwent a second DXA analysis after 18 months of follow-up. Considering the mean age of patients enrolled in the study, bone densitometry at the lumbar spine was not included in the study because of potential artefacts associated with degenerative diseases.

### 2.4. Statistical Analysis

Data are expressed throughout as means ± standard deviation. Differences between groups were assessed using a Student’s *t*-test for paired and unpaired data. Multiple linear regression analysis was performed to evaluate the potential relationship between variables. Statistical significance was assumed when *p*-values were less than 0.05. Statistical analysis was carried out using SPSS for Windows version 22.0 (SPSS Inc.; Chicago, IL, USA). 

## 3. Results

### 3.1. Groups of Treatment: Daily vs. Monthly

The anthropometric and biochemical variables of both groups are reported in Table 1. The two group did not differ for any parameter evaluated, especially with regards to the concentrations of serum 25(OH) D. The mean BMI of patients was 27, indicating the presence of overweight in the population enrolled in the study. 

The medium daily calcium intake evaluated by the questionnaire was 907 mg/day.

After six months, in both groups, we observed a significant increase in serum calcium concentration, which remained in the normal range. We also observed a significant increase of 25(OH) D concentration, which reached the normal range in almost all patients. The 25(OH) D concentration was lower than 30 ng/mL in only eight patients after six months, and those eight patients had the highest BMI. In the first group, we observed a mean increase of 14 ng/mL, while, in the second group, the increase was about 13 ng/mL. 

At baseline, we found a linear inverse correlation among BMI, WtHR, waist circumference, and 25(OH) D concentrations (Figure 2).

On the basis of this result, we performed a linear regression to evaluate the main determinant of vitamin D concentration at six months. Waist circumference was the main determinant after corrections for age, baseline level of vitamin D, and calcium intake (*R* = 0.35; *p* < 0.05).

### 3.2. Follow-Up Study of Monthly Treatment

After 18 months of treatment with a single monthly dose of 50,000 UI of cholecalciferol, all patients completed the study, and compliance with supplementation was 100%.

Regarding biochemical variables, we observed a significant increase in calcium level, associated with a decrease in serum PTH levels and bone turnover, while 25(OH) D concentrations were maintained, despite an insignificant increase (Figure 3).

Regarding anthropometric variables, we observed a significant decrease in waist circumference. The waist circumference at 18 months was also significantly lower compared with that at six months (102 cm, *p* < 0.05; Figure 4). 

Bone density did not change at the end of the study. 

Furthermore, after 18 months, waist circumference was the main determinant of 25(OH) D concentration, evaluated by linear regression after corrections for age, bone turnover, and calcium intake. The diet of patients did not change during the study period; however, all patients increased their physical activity, as suggested at baseline.

## 4. Discussion

Activated hormonal vitamin D3 (calcitriol), by binding to the vitamin D receptor (VDR), induces several biological and physiological processes [17]. The importance of vitamin D in pregnancy is well known [18], and experimental animal studies support an active contribution of vitamin D to organ development [19,20]. Quite recently, Eggemoen et al. demonstrated that vitamin D deficiency in early pregnancy was strongly associated with birth weight and measures of neonatal body composition, when adjusted for gestational age, neonate gender, maternal age, education, and pre-pregnancy BMI [21]. However, after including ethnicity in the model, maternal 25(OH) D during pregnancy was no longer associated with birth weight [21]. On the other hand, optimal serum levels in postmenopausal women could be fundamental for reducing the number falls and preserving muscle function, as well as reducing the occurrence of respiratory infections and colorectal adenomas [22,23]. The complex of ligand-bound VDR with the retinoid-X receptor (RXR) is able to influence gene expression by targeting the response element in the promoter region of specific genes [5]; in addition, calcitriol can activate the intracellular pathways in a manner independent of nuclear VDR binding [17]. It was shown that vitamin D deficiency and obesity are strongly correlated [24], and low serum levels of vitamin D in obese population were observed [25]. In fact, due to the lipophilic characteristic, the vitamin D is stored and then retained by the adipose tissue [26] and, consequently, 25(OH) D appears to dilute into both the circulation and fat stores. However, it was reported that increases in BMI were associated with reductions in serum 25(OH) D; however, variation of serum 25(OH) D was not associated with any changes in body fat/BMI [27]. It is known that adipose tissue arises from the maturation process of pre-adipocyte cells and that these cells are copious in obese individuals [28,29]. Interestingly, it was demonstrated, at least in early life and in children, that pre-adipocytes are an important target cells of 1,25(OH)_2_D3, and that, in particular, this hormone induces growth arrest in pre-adipocytes. On the contrary, 1,25(OH)_2_D3 has less effects in mature adipocytes [30], and this finding could be due to the reduced expression of VDR in mature adipocytes [28]. However, it was also demonstrated that 1,25(OH)_2_D3 inhibits adipocyte differentiation in a dose-dependent manner. In particular, 1,25(OH)_2_D3 inhibits adipogenesis by reducing the expression of genes coding for CCAAT-enhancer-binding protein (C/EBP), PPARγ, lipoprotein lipase (LPL), and adipocyte protein 2 (aP2), involved in the early and late phase of adipocyte differentiation [31]. Importantly, Kong et al. demonstrated that the downregulation of PPARy and C/EBP expression by 1,25(OH)_2_D3 is a primary and direct event in blocking adipogenesis; therefore, 1,25(OH)_2_D3 acts by affecting the transcription factors that are important players in adipocyte differentiation. 

Despite all the potential contributions of vitamin D to human health, there is no consensus regarding dosage and time/interval period of administration [2,32]. The factors influencing dose response to vitamin D supplementation (i.e., baseline vitamin D level, gender, ethnicity, body composition, calcium/phosphorus status, and genes) are well known, although the requirements for daily vs. weekly or monthly administration are still under debate [33]. Our results, at the group level, after six months of vitamin D administration either daily or monthly, showed a significant increase in vitamin D status that reached the normal range in almost all patients. The six-month period was selected according to suggestions in literature regarding this time period as optimal for monitoring vitamin D status [34].

However, we need to underline that the eight patients with lower levels of vitamin D than 30 ng/mL after six months were obese. Keeping this finding in mind, we performed linear regression in order to evaluate the main determinant of vitamin D level at six months, controlling for all possible confounding factors for calcium intake in the normal range. Waist circumference was the main determinant, and was definitely correlated with the relationship between vitamin D absorption and fat mass. Therefore, we could speculate that the detriment of adipocytes, evaluated by means of waist circumference, can drive the differentiation of mesenchymal stem cells into osteoblasts. The influence of vitamin D on the commitment of mesenchymal stem cells could explain the decline in bone densitometry after 18 months in old subjects, which was unchanged in our patients during the observation time. In addition, it is known that the transcription factor PPARγ and the transcription factor RUNX2 control adipogenesis and osteogenesis by inducing the differentiation of mesenchymal stem cells into adipocytes or osteoblasts, respectively. In particular, the relationship between mesenchymal progenitors and osteoblastogenesis or adipogenesis is determined by the competition between RUNX2 and PPARγ [8]. However, the reduced waist circumference that we observed in patients treated with vitamin D could be explained by the direct effect of 1,25(OH)_2_D3 on the intracellular pathway of preadipocytes. This finding is important, as the waist circumference, which reflects visceral adiposity, is directly correlated with cardiovascular diseases and metabolic disturbances, showing clearly that serum 25(OH) D is dependent on body mass [35,36]. We cannot forget the concept of storage of vitamin D in the adipose tissue, and this finding can potentially confirm its causative role.

Our study had some limitations. The sample size could be deemed questionable. However, the results are clear due to the decision to follow the subjects undergoing monthly treatment for a long period of time. Other variables could potentially have been collected; however, we decided to concentrate our analyses on dosage time and results related to mass. 

Considering our findings and those in the recent literature [2,37,38,39], it seems that the importance of vitamin D for general health is becoming increasingly apparent [23].

## Figures and Tables

**Figure 1 nutrients-10-01934-f001:**
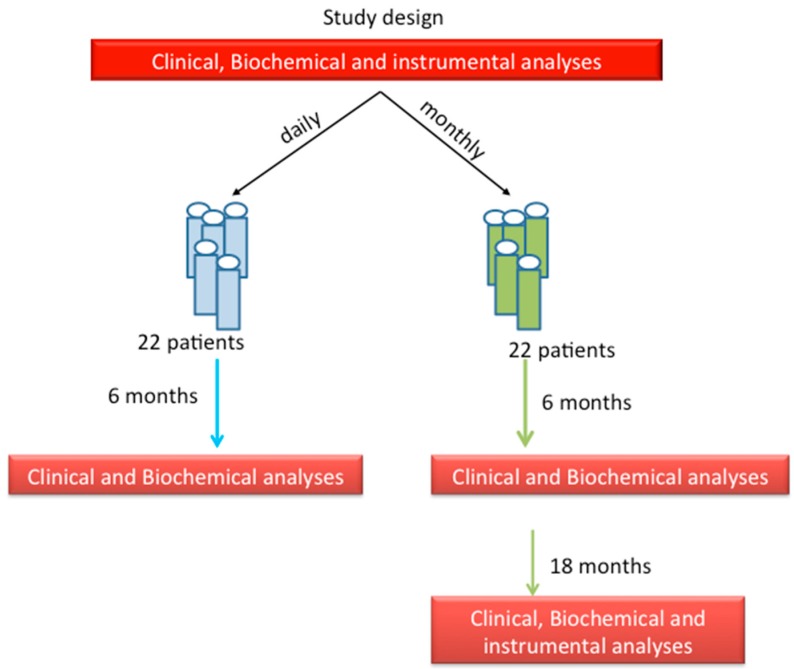
Description of study design.

**Figure 2 nutrients-10-01934-f002:**
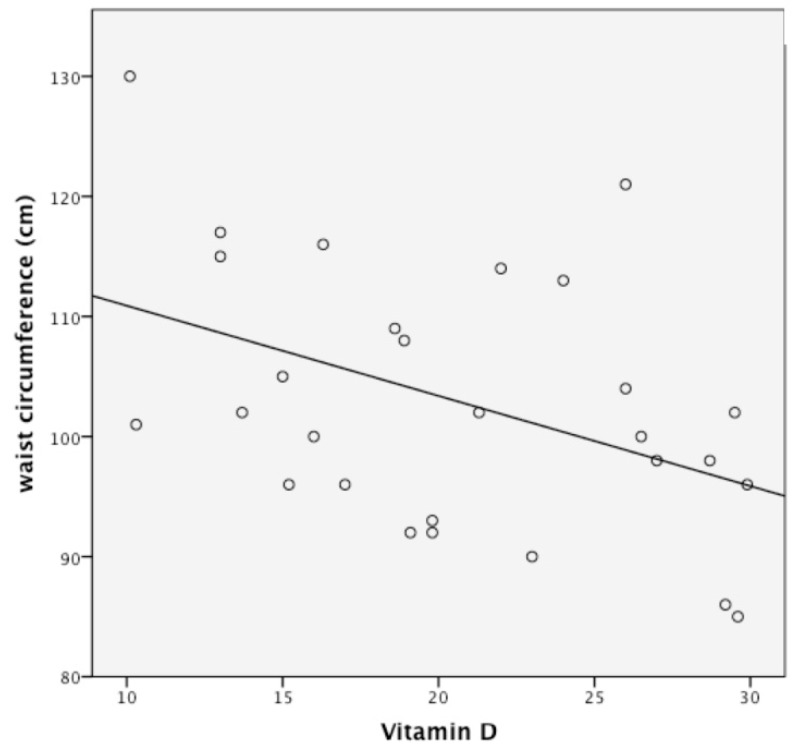
Linear correlation between serum 1,25(OH)_2_D (25(OH) D) and waist circumference at baseline; *p* < 0.05; *R* = 0.35.

**Figure 3 nutrients-10-01934-f003:**
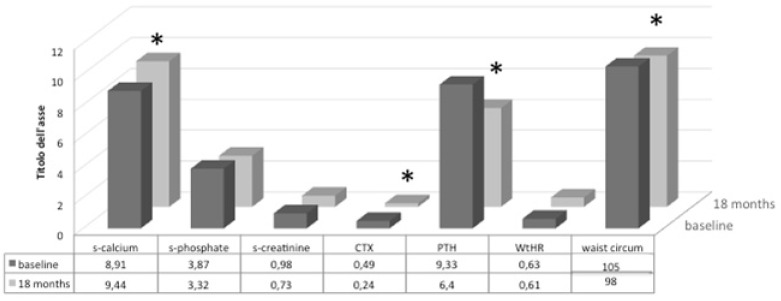
Main anthropometric and biochemical variables of monthly group at baseline and after 18 months of treatment; * *p* < 0.05.

**Figure 4 nutrients-10-01934-f004:**
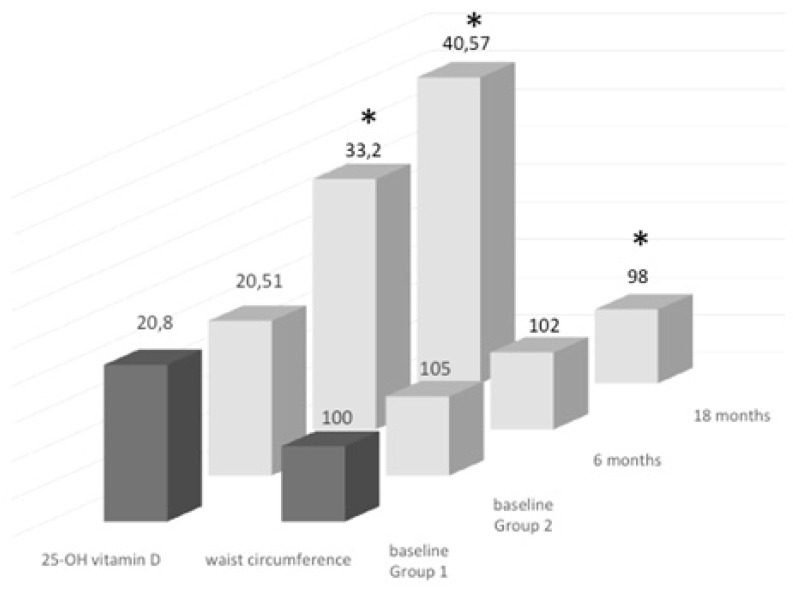
Serum 25(OH) D levels and waist circumference measurements at baseline in both groups of treatment, and variations along the 18-month study follow-up; * *p* < 0.05.

**Table 1 nutrients-10-01934-t001:** Anthropometric, densitometric, and biochemical variables of the study population undergoing the two regimens of treatment.

Parameter	Group 1 (Daily)	Group 2 (Montly)	*P*
Number of subjects	22	22	NS
Sex (M/F)	10/12	9/13	NS
Age (yrs)	77 ± 9	78 ± 8	NS
Weight (kg)	76 ± 18.6	77 ± 11.2	NS
BMI (kg/m2)	27.7 ± 6.5	27.3 ± 4	NS
Waist circumference (cm)	99.5 ± 13	105 ± 9	NS
WtHR	0.60 ± 0.78	0.63 ± 0.07	NS
25(OH) vitamin D (ng/mL)	20.81 ± 6.98	20.51 ± 5.61	NS
s-Calcium (mg/dL)	8.99 ± 0.45	8.91 ± 0.29	NS
s-Phoshate (mg/dL)	2.52 ± 0.67	3.87 ± 0.56	NS
s-Creatinine (mg/dL)	0.96 ± 0.23	0.98 ± 0.25	NS
CTX (ng/mL)	0.50 ± 0.37	0.49 ± 0.31	NS
PTH (pmol/L)	8.39 ± 3.31	9.33 ± 6.76	NS
ALP (U/L)	85.25 ± 31.18	100.5 ± 74.24	NS
Total Hip BMD (g/cm^2^)	0.821 ± 0.105	0.873 ± 0.101	NS
Total Hip T-score (SD)	−1.0 ± 0.9	−0.3 ± 0.7	NS
Femoral Neck BMD (g/cm^2^)	0.656 ± 0.137	0.798 ± 0.201	NS
Femoral Neck T-score (SD)	−1.7 ± 0.3	−0.7 ± 0.7	NS

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
