# Peer review of "Vitamin D Daily versus Monthly Administration: Bone Turnover and Adipose Tissue Influences"

_nutrients, 2018, doi:10.3390/nu10121934_

Round 1
Reviewer 1 Report
1) Was there a difference in wait circumference from 6 month to 18 month in monthly dosed group?
2) Has there been any lifestyle changes (diet, exercise) during the study period?
3) Another figure summarizing the findings showing a bar diagram of Group 1 with baseline Vit. D level with waist circumference and group 2 with 6th month and 18th month Vit. D level with their wait circumference would be informative.
Author Response
We thank reviewer 1 for his positive reaction and or giving us the suggestion of a figure showing the relationship between Vit. D level and waist circumference along the entire study design (Group 1 Vs Group 2; 6 months Vs 18 months)
1) Was there a difference in wait circumference from 6 month to 18 month in monthly dosed group?
As requested, we have added the information regarding waist circumference difference.
2) Has there been any lifestyle changes (diet, exercise) during the study period?
We have added the information regarding life style.
3) Another figure summarizing the findings showing a bar diagram of Group 1 with baseline Vit. D level with waist circumference and group 2 with 6th month and 18th month Vit. D level with their wait circumference would be informative.
Once again we thank the reviewer for this nice and brilliant suggestion and we have done the figure according with reviewer suggestion.

Reviewer 2 Report
Review of MS:- Nutrients-VitD-Monthly-vs-daily by Carbonare et al.
This MS reports a comparison of the effects of supplemental oral vitamin D at 1750 IU/day and at 50,000 IU monthly in a small group of older adults with baseline vitamin D deficiency and reports equivalent status in these two groups after 6 months and that after 18 months of interval supplementation serum calcium had increased and both waist circumference and bone turnover were significantly lower than at baseline. The text can be followed, but with some difficulty in places and editing of the English text by a native English speaker is required for improving the readability of the MS and also the clarity of the science in various parts of the MS as mentioned in the specific comments where an effort has been made to suggest how the English usage might be improved.
General comments.
1. the MS uses the term ‘vitamin D’ to cover many different things but this term is not appropriate for many of them; thus, many amendments and clarifications are needed in the text, to provide accuracy and clarity in the report, as follows:-
1a). The ‘vitamin D given as a supplementation is not always specified though it was vitamin D3 that was given.
1b) Vitamin D was not measured in the serum, but the 25(OH)D metabolite; whether the assay used was sensitive to both the D2 and D3 forms of this metabolite is not mentioned but should be specified. It is also necessary to report the exact assay used to measure serum 25(OH)D, to report the CVs of that assay, and the compliance of that assay system with whatever external quality assurance scheme was used to validate the assay runs providing the 25(OH)D data since those assays are very variable and methodology has changed greatly over time. This information will allow readers to consider the likely accuracy and specificity of the data.
1c). Laboratory measurements are made as concentrations in body fluids, not levels, and all the assay data, including that for serum 25(OH)D should, therefore, be reported as concentrations, not levels; the term levels is however useful for discussing cut-offs.
1d). Much of the literature on vitamin D modulating the maturation of pre-adipocytes towards muscle rather than into adipocytes is in early life and in children as I remember it, so please check that this is reported in adults and adjust the text if that is not the case.
Specific comments by section and line number. These cover both matters of content where clarification would be useful and in addition, those matters of language where some unusual wording interferes with the readers ability to follow the text at a first reading.
Abstract. Line 16, dosage or route of administration……; aims of our study were ….; line 18-19, the effects of different routes of administration of ……… …..with low levels of serum 25(OH)D at baseline.; line 20, ..with hypovitaminosis D at baseline….; line 27, say what turnover was decreased – [I think you mean bone turnover]; line 28, I think you mean, support the efficacy of vitamin D3 given monthly both for correcting …...; line 30, ….between serum 25(OH)D concentration and waist circumference which, since waist size is directly associated with the risk of …….., supports vitamin D having a protective role in the current setting.
Introduction. Line 36, ……..its hormonally active metabolite…… are…. Line 40, …. gene promoting osteogenic ….?; line 47-8, is this effect reported in adults, [please check as I have only met reports of this in early life and young children]; line 49, hormonal vitamin D?; line 51 is an example where serum concentration would be better than level; line 52, not ‘would involve’, but ‘should be’; line 59/60, ‘the roles of vitamin D in energy metabolism are well known’ [so give some examples with reference[s] for those new to this field]; line 69, ‘aims of’ line 72, say either low vitamin D status or low baseline concentrations of serum 25(OH)D….; line 74, for maintaining….line 78, visits to….
Patients and methods. If these were volunteers, were they healthy subject or attending with medical conditions, I see the exclusions but how did they come to be patients?
Figure 1, showing the study design should include the baseline assessments; line 89, naïve for consumptionm9f vitamin D supplements [surely not metabolites as you did not supplement with metabolites but with intact cholecalciferol. ; line 94, vitamin d was not measured so say vitamin D status; line 95, ..as well as any condition affecting ….line 97, is that biohumoral? Line 100, …and we maintained supplementation [not treatment] for …..; line101, …same evaluations as at baseline; line 103, omit ‘to’ before ‘the following’; lines 111+ were the anthropometric measurements made at a standard time of day and with standardised equipment?; line 113-4, was dietary intake of vitamin D [2/3] assessed, and if so did it or did it not contribute to the determination of serum concentrations of either serum calcium or serum 25(OH)D 2/3 concentrations? line 116, at the proximal….; line 119, …patients on monthly treatment…; line 121, …at the lumbar spine….; line 124 was the data collected all normally distributed – if not, medians should be used for non-normally distributed data; line 126, was this multiple regression analysis [please specify]; line 125, paired is the more usual term rather than coupled.
Results. Line 131 and 132, things measured and assessed as part of a study are variables and not parameters. [Please check this throughout the text and Figures and the heading of Table 1]; line 133, ..the concentrations of serum 25(OH)D were similar.; line 139, surely that calcium intake is the recommended daily intake [preferably from the diet]; line 140, in serum calcium concentration…; line 142,and lines 151 and 152, ………………. 25(OH)D concentration, as vitamin D was not measured, etc, as in general comments above]; line 143-4, ..lower then 30ng/ml after 6 months and those 8 patients had the highest BMIs; line 146 linear associations, but state that they were inverse; line 147, concentration not level as also in the Legend to Figure 2 where the labelling ;’vitamin D should be changed to serum 25(OH)D; line 153, if p was<0.05 why not state what is actually , line 156+ , ….and compliance with supplementation was 100%; line 158, variables not parameters; line 159, ..decreases of both serum PTH and of bone turnover while serum 25(OH)D concentrations were …. Figure 3 ensure that the data shown is explained, [i.e. show the shading for the two bar columns and say which is which. Also,. It is, again, variables and not parameters both here and in line 164; line 166, 25(OH)D concentrations, [yet again].
Discussion. Line 170, vitamin D is inert, so ‘Activated hormonal vitamin D [calcitriol] binding to …; line 171-2, ‘The importance of vitamin D in pregnancy is well known ….[note that recent papers show this is probably not true for fetal bone, so consider saying what it is important for, such as perhaps deny this for bone health, consider saying what it is important for, [maybe pre-eclampsia and gestational diabetes?]; line 175, …the complex of the ligand-bound VDR with the ….; RXR …; line 177, …addition, calcitriol can activate ……..in a manner independent of the nuclear VDR…..lines 181-3, on obesity and serum 25(OH)D, intact vitamin D is stored in fat, but 25(OH)D appears to dilute into both the circulation and fat stores, see two-way Mendelian Randomisation study reported by the group with Hypponen E, that shows increases in BMI are associated with falls in serum 25(OH)D but that variation of serum 25(OH)D is not associated with any changes in body fat/BMI [that reference may be useful]; lines 185-190, remove the word ‘the’ where redundant in that paragraph; line 195, in adipocyte differentiation; line 201, under debate; line 202 and 205 use ‘vitamin D status’ rather than vitamin D; line 214+, ..why …known to decline over 18 months in older people, was unchanged in out subjects over that period,’; line 220, should ‘completion’, read ‘competition’? Please clarify; line 223, that reflects visceral adiposity…; line 225-6, that serum 25(OH)D was dependent on body size [as already mentioned] and you might mention the work of Ekwaru et al who found that serum 25(OH)D rose less in subjects given different sized supplements, the higher their BMI. Line 226-7 needs to be rephrased as it is not thought not be anything to do with intact vitamin D storage in fat, as referred to above. line 232, body mass or bone mass or both? Line 233-4, reads with difficulty, do you mean something like ‘considering our findings and those in the recent literature it seems that the importance of vitamin D for general health is becoming increasingly apparent’
Author Response
Reviewer 2
This MS reports a comparison of the effects of supplemental oral vitamin D at 1750 IU/day and at 50,000 IU monthly in a small group of older adults with baseline vitamin D deficiency and reports equivalent status in these two groups after 6 months and that after 18 months of interval supplementation serum calcium had increased and both waist circumference and bone turnover were significantly lower than at baseline. The text can be followed, but with some difficulty in places and editing of the English text by a native English speaker is required for improving the readability of the MS and also the clarity of the science in various parts of the MS as mentioned in the specific comments where an effort has been made to suggest how the English usage might be improved.
We thank the reviewer 2 for his/her useful comments. We have added all the suggestions made and we have done also all the correction where requested.
General comments.
1. the MS uses the term ‘vitamin D’ to cover many different things but this term is not appropriate for many of them; thus, many amendments and clarifications are needed in the text, to provide accuracy and clarity in the report, as follows:-
1a). The ‘vitamin D given as a supplementation is not always specified though it was vitamin D3 that was given.
We specified the supplementation as Vitamin D3 in the text.
1b) Vitamin D was not measured in the serum, but the 25(OH)D metabolite; whether the assay used was sensitive to both the D2 and D3 forms of this metabolite is not mentioned but should be specified. It is also necessary to report the exact assay used to measure serum 25(OH)D, to report the CVs of that assay, and the compliance of that assay system with whatever external quality assurance scheme was used to validate the assay runs providing the 25(OH)D data since those assays are very variable and methodology has changed greatly over time. This information will allow readers to consider the likely accuracy and specificity of the data.
We specified the characteristics of the assay in the method session.
1c). Laboratory measurements are made as concentrations in body fluids, not levels, and all the assay data, including that for serum 25(OH)D should, therefore, be reported as concentrations, not levels; the term levels is however useful for discussing cut-offs.
We corrected the terms, as suggested.
1d). Much of the literature on vitamin D modulating the maturation of pre-adipocytes towards muscle rather than into adipocytes is in early life and in children as I remember it, so please check that this is reported in adults and adjust the text if that is not the case.
We cannot find any information in adults. So, we specified in the text that this aspect was demonstrated in early life and children.
Specific comments by section and line number. These cover both matters of content where clarification would be useful and in addition, those matters of language where some unusual wording interferes with the readers ability to follow the text at a first reading.
Abstract. Line 16, dosage or route of administration……; aims of our study were ….; line 18-19, the effects of different routes of administration of ……… …..with low levels of serum 25(OH)D at baseline.; line 20, ..with hypovitaminosis D at baseline….; line 27, say what turnover was decreased – [I think you mean bone turnover]; line 28, I think you mean, support the efficacy of vitamin D3 given monthly both for correcting …...; line 30, ….between serum 25(OH)D concentration and waist circumference which, since waist size is directly associated with the risk of …….., supports vitamin D having a protective role in the current setting.
Introduction. Line 36, ……..its hormonally active metabolite…… are…. Line 40, …. gene promoting osteogenic ….?; line 47-8, is this effect reported in adults, [please check as I have only met reports of this in early life and young children]; line 49, hormonal vitamin D?; line 51is an example where serum concentration would be better than level; line 52, not ‘would involve’, but ‘should be’; line 59/60, ‘the roles of vitamin D in energy metabolism are well known’ [so give some examples with reference[s] for those new to this field]; line 69, ‘aims of’ line 72, say either low vitamin D status or low baseline concentrations of serum 25(OH)D….; line 74, for maintaining….line 78, visits to….
We made all the suggested corrections.
Patients and methods. If these were volunteers, were they healthy subject or attending with medical conditions, I see the exclusions but how did they come to be patients?
Patients were enrolled during a routine visit aimed to evaluate the possible presence of a metabolic bone disease. We specified in the text this aspect.
Figure 1, showing the study design should include the baseline assessments;
We updated figure 1, as suggested.
line 89, naïve for consumptionm9f vitamin D supplements [surely not metabolites as you did not supplement with metabolites but with intact cholecalciferol. ; line 94, vitamin d was not measured so say vitamin D status; line 95, ..as well as any condition affecting ….line 97, is that biohumoral? Line 100, …and we maintained supplementation [not treatment] for …..; line101, …same evaluations as at baseline; line 103, omit ‘to’ before ‘the following’; lines 111+ were the anthropometric measurements made at a standard time of day and with standardised equipment?;
We corrected and specified these points in the text.
line 113-4, was dietary intake of vitamin D [2/3] assessed, and if so did it or did it not contribute to the determination of serum concentrations of either serum calcium or serum 25(OH)D 2/3 concentrations?
The intake of vitamin D2/D3 in patients enrolled in the study was negligible, thus we reported only the calcium intake.
line 116, at the proximal….; line 119, …patients on monthly treatment…; line 121, …at the lumbar spine….; line 124 was the data collected all normally distributed – if not, medians should be used for non-normally distributed data;
Data was normally distributed, thus we maintained the results expressed as means.
line 126, was this multiple regression analysis [please specify]; line 125, paired iss the more usual term rather than coupled.
We corrected and specified statistical description, as suggested.
Results. Line 131 and 132, things measured and assessed as part of a study are variables and not parameters. [Please check this throughout the text and Figures and the heading of Table 1]; line 133, ..the concentrations of serum 25(OH)D were similar.; line 139, surely that calcium intake is the recommended daily intake [preferably from the diet]; line 140, in serum calcium concentration…; line 142,and lines 151 and 152, ………………. 25(OH)D concentration, as vitamin D was not measured, etc, as in general comments above]; line 143-4, ..lower then 30ng/ml after 6 months and those 8 patients had the highest BMIs; line 146 linear associations, but state that they were inverse; line 147, concentration not level as also in the Legend to Figure 2 where the labelling ;’vitamin D should be changed to serum 25(OH)D; line 153, if p was<0.05 why not state what is actually , line 156+ , ….and compliance with supplementation was 100%; line 158, variables not parameters; line 159, ..decreases of both serum PTH and of bone turnover while serum 25(OH)D concentrations were …. Figure 3 ensure that the data shown is explained, [i.e. show the shading for the two bar columns and say which is which. Also,. It is, again, variables and not parameters both here and in line 164; line 166, 25(OH)D concentrations, [yet again].
We made the suggested corrections and we changed the labels of figure 3 for a better understanding.
Discussion. Line 170, vitamin D is inert, so ‘Activated hormonal vitamin D [calcitriol] binding to …; line 171-2, ‘The importance of vitamin D in pregnancy is well known ….[note that recent papers show this is probably not true for fetal bone, so consider saying what it is important for, such as perhaps deny this for bone health, consider saying what it is important for, [maybe pre-eclampsia and gestational diabetes?];
As the reviewer pointed out, we rephrased the concept and we added references that could support our statement.
line 175, …the complex of the ligand-bound VDR with the ….; RXR …; line 177, …addition, calcitriol can activate ……..in a manner independent of the nuclear VDR…..lines 181-3, on obesity and serum 25(OH)D, intact vitamin D is stored in fat, but 25(OH)D appears to dilute into both the circulation and fat stores, see two-way Mendelian Randomisation study reported by the group with Hypponen E, that shows increases in BMI are associated with falls in serum 25(OH)D but that variation of serum 25(OH)D is not associated with any changes in body fat/BMI [that reference may be useful]; lines 185-190, remove the word ‘the’ where redundant in that paragraph; line 195, in adipocyte differentiation; line 201, under debate; line 202 and 205 use ‘vitamin D status’ rather than vitamin D; line 214+, ..why …known to decline over 18 months in older people, was unchanged in out subjects over that period,’; line 220, should ‘completion’, read ‘competition’? Please clarify; line 223, that reflects visceral adiposity…; line 225-6, that serum 25(OH)D was dependent on body size [as already mentioned] and you might mention the work of Ekwaru et al who found that serum 25(OH)D rose less in subjects given different sized supplements, the higher their BMI.
Line 226-7 needs to be rephrased as it is not thought not be anything to do with intact vitamin D storage in fat, as referred to above.
As the reviewer suggested we have fixed and clarify the sentence.
line 232, body mass or bone mass or both? Line 233-4, reads with difficulty, do you mean something like ‘considering our findings and those in the recent literature it seems that the importance of vitamin D for general health is becoming increasingly apparent’
We corrected the sentences, as suggested.
